# Evolutionary radiation strategy revealed in the Scarabaeidae with evidence of continuous spatiotemporal morphology and phylogenesis

Yijie Tong [1], Yuanyuan Lu [1], Zhehao Tian[1,2], Xingke Yang[1,3] & Ming Bai [1,4,5,6] ✉

Evolutionary biology faces the important challenge of determining how to interpret the relationship between selection pressures and evolutionary radiation. The lack of morphological evidence on cross-species research adds to difficulty of this challenge. We proposed a new paradigm for evaluating the evolution of branches through changes in characters on continuous spatiotemporal scales, for better interpreting the impact of biotic/abiotic drivers on the evolutionary radiation. It reveals a causal link between morphological changes and selective pressures: consistent deformation signals for all tested characters on timeline, which provided strong support for the evolutionary hypothesis of relationship between scarabs and biotic/abiotic drivers; the evolutionary strategies under niche differentiation, which were manifested in the responsiveness degree of functional morphological characters with different selection pressure. This morphological information-driven integrative approach sheds light on the mechanism of macroevolution under different selection pressures and is applicable to more biodiversity research.

Evolutionary radiation events lead to broad cladogenesis, rapid divergence, and adaptive evolution between lineages[1,2]. Determining how diverse organisms respond to abiotic and biotic pressures is a significant challenge for understanding evolution[3]. Different organisms have amassed extraordinary species richness in response to selection pressures, including a wide range of morphological diversity[4], which is essential to comprehending their evolutionary strategies[5]. To this end, microevolutionary studies aim to explore the mechanisms of phenotypic changes under various selection pressures through synthesis and protein screening[6–10]. However, due to limitations of species-specificity and time scales in microevolutionary studies, inferring the phylogenetic relationships of the major living species and their evolutionary mechanisms is unachievable to a certain extent. Macroevolutionary studies extract genetic information from the gene pool of a greater variety, then combine existing species morphology and recorded fossil information to construct phylogenetic relationships between higher orders[11,12], thus revealing the

relationship between evolutionary radiations and selection pressures (temperature, reciprocity, etc.)[13]. Nevertheless, the qualitative morphological methods and limited morphological information coverage widely used in cross-species research lead to ambiguity regarding the mechanisms driving diversity[14,15]. Harmon analyzed the characters of lizards at the stages of divergence event by quantitative means, and revealed interspecific competition and ecological release in the early history of evolution[1]. However, due to the clades' erratic trajectories and various selection pressures during sequential historical differentiation[16,17], the macroevolutionary models of taxa remain controversial without morphological data on continuous time scales[18,19]. In the process of radiation evolution of biological groups, the relationship between morphological changes and the effects of selection pressures has not been well understood.

In light of the different drivers that may affect the evolutionary history of a given clade, the evolutionary model of biological taxa is probably best

[1]Key Laboratory of Animal Biodiversity Conservation and Integrated Pest Management (Chinese Academy of Sciences), Institute of Zoology, Chinese Academy of Sciences, Beijing 100101, China. [2]School of Agriculture, Ningxia University, Yinchuan 750021, China. [3]Guangdong Key Laboratory of Animal Conservation and Resource Utilization, Guangdong Public Laboratory of Wild Animal Conservation and Utilization, Institute of Zoology, Guangdong Academy of Sciences, Guangzhou 510260, China. [4]College of Plant Protection, Hebei Agricultural University, Baoding 071001, China. [5]Northeast Asia Biodiversity Research Center, Northeast Forestry University, Harbin 150040, China. [6]College of Life Sciences, University of Chinese Academy of Sciences, Beijing 100049, China. ✉e-mail: baim@ioz.ac.cn

determined by studying individual cases. There are ~27,000 species of 1600 genera in Scarabaeidae worldwide[20], which have evolved three typical feeding types (omnivory, coprophagy and phytophagy) and some special feeding types (e.g., predatory, necrophagy) via coevolution under strong selection pressures[5]. The coprophagous and phytophagous scarabs have successfully undergone evolutionary radiation, and the former group, which is represented by Aphodiinae and Scarabaeinae, has rapidly differentiated into ~8800 species by occupying distinct ecological niches with specific feeding types[21,22]. The extensive species richness and divergent morphology between feeding-type groups make scarabs a dominant group for studying biodiversity evolution[23]. In-depth biological and morphological research has shed light on the phylogenetic relationships and morphological adaptive evolution of Scarabaeidae[24], but it was not until the advent of molecular biology that research on the pace of biological differentiation and evolutionary hypothesis testing accelerated[25]. The relevant research speculates on the differentiation at ancestral nodes, and interpreting the macroscopic changes in scarabs in a way[26,27]. Bai found that coprophages and phytophages originated from omnivorous ancestors, revealing the morphological transformation of mandibles for handling various food resources[5], and Ahrens revealed significant diversity connections between scarabs and other ecological joiners (mammals and angiosperms) that were driven by food specialization[28]. However, phylogenetic relationships and fragmentary fossil evidence make it difficult to assess the clade evolution of Scarabaeidae, and the single test feature and low coverage of morphological information cannot better interpret their differentiation and evolutionary radiation under selection pressures.

In this study, the historical variations in the functional morphological characters of scarabs with various feeding types were used as examples. We inferred the morphological differences between all historical branch nodes of Scarabaeidae by merging geometric morphometrics and phylogenetics, we then interpret the impact of biotic/abiotic driving factors on the evolutionary radiation of Scarabaeidae through a new paradigm for evaluating the evolution of branches through changes in characters on continuous spatiotemporal scales. Three analyses were carried out: (1) we analyzed the morphological diversity of different characters of scarabs with different feeding types; (2) we were able to determine the deformation ratio index (DR) by calculating the morphological differences between successive branch nodes on the time scale, for exploring the correlation between changes in morphology–shifts in the deformation rate (DR) metric–with changes in selective pressures, which in turn are correlated with abiotic drivers such as global temperature; and (3) we processed the DR of different characters and performed horizontal comparison at the same spatiotemporal scale, for analyzing the changes of features with different function under selection pressure and interpreting the radiation evolution strategy. This study reveals differentiation events of scarabs in the Upper Jurassic and Cretaceous that were influenced by global temperature, mammals, and angiosperms: (1) we found extremely consistent deformation signals for all tested characters on the timeline, which provided strong support for the evolutionary hypothesis of the relationship between coprophagous scarabs and mammals; (2) we proposed the important role of the elytra rather than long-considered mandibles in the niche differentiation of coprophages during the Cretaceous[5], which were manifested in the degree of responsiveness of different functional morphological characters under the influence of selection pressure. Furthermore, this study explained the gradual evolution of organisms from a quantitative morphological point of view: our results showed that scarabs did not suddenly become the dominant taxa with the rise of angiosperms in the Palaeogene but began to respond to the occurrence of angiosperms in the Cretaceous, responses that manifested as small morphological changes in characters. This study, which uses a set of continuous spatiotemporal morphological data, expands the coverage of morphological data for examining taxon evolution at a spatiotemporal scale through an integrated methodology. It interprets the macroevolutionary model by comparing the diversification of different characters, offering a fresh perspective for detailed studies of the selection pressures and formation mechanisms regulating the rise and fall of species.

## Results

### Character morphological diversity in different feeding-type groups

Based on the PCA plot (PC1-PC2), the morphological diversity of different characters among the test feeding types was shown. We observed noticeable variation between coprophages and phytophages, especially in the mandible and hindwing tests (Fig. 1a). More precisely, the morphological differences in the mandible were mainly concentrated in the angle between the incisor lobe and molar lobe, which changed from an obtuse angle to a right angle (Fig. 1a, mandible pattern, PC2), and the distance between the end of RA3 and RA44 of the hindwing varied from long in coprophagous species to short in phytophagous species (Fig. 1a, hindwing pattern, PC1). The deformations of the pronotum and elytron were mainly concentrated in the anterior angle, the posterior angle, and the axial aspect ratio of the main body parts (Fig. 1a, pronotum/elytron pattern, PC1-2). Based on the character differences revealed by PCA, CMA confirmed that the morphology of the test characters was significantly different (Supplementary Table 1), and the total estimated correctness of discrimination between feeding-type groups was 97.60%/59.10%/89.06%/92.78% for the mandible/pronotum/elytron/hindwing test, respectively.

### Feeding types of major ancestral lineages of Scarabaeidae

Based on the estimation results of the ancestral feeding type with a mean divergence time in Scarabaeidae (Fig. 1b) (Supplementary Table 2), we found that the ancestors of Scarabaeidae differentiated their feeding types from omnivory to coprophagy and phytophagy at 137 Ma (Node 14, 95% CIs from 89 to 156 Ma), and they retained the omnivorous feeding type of Rhyparini (Node 1, 95% CIs from 66 to 126 Ma). Then, the coprophagous lineage differentiated into the subfamilies Aphodiini and Scarabaeinae at 93 Ma (Node 11, 95% CIs from 58 to 115 Ma), and the phytophagous lineage gradually diverged into Cetoniinae/Dynastinae/Melolonthinae/Rutelinae in the Cretaceous.

### DR's changing trend among major lineages of Scarabaeidae

The step line charts of the DR for the test characters were analyzed (Fig. 2b–e) (Supplementary Data 1; Supplementary Table 3). Based on the data obtained with the mean divergence time of Scarabaeidae, the ancestral node's DR of the mandible gradually increased at 170 Ma (Node 17, 95% CIs from 111 to 195 Ma), and it reached the highest peak at 156 Ma (Node 16, 95% CIs from 102 to 178 Ma); then, it reached the second peak and the third peak in Melolonthini at 102/85 Ma, respectively. In the test of the pronotum/hindwing, the ancestral node's DR gradually increased at 156 Ma (Node 16, 95% CIs from 102 to 178 Ma), reached the first peak at 146 Ma (Node 15, 95% CIs from 95 to 166 Ma), and then reached the second peak (also the highest peak in the hindwing test) at 102 Ma (Node 13, 95% CIs from 66 to 122 Ma). It decreased from the mean divergence time of 95 Ma to modern times, although Melolonthini reached the third peak at 85 Ma (in the pronotum test). Through morphological diversity analysis of the elytron, we found that the ancestral node's DR reached the first peak at 146 Ma and the second peak at 105/102 Ma for coprophages/phytophages (Euchirini), respectively. Phytophagy subsequently differentiated the highest peak in the lineage of Hopliini/Melolonthini at 85 Ma.

### Characters' SGR in the test lineages of Scarabaeidae

The DR trends of characters in the different feeding-type groups were analyzed and compared horizontally by the lines of best fit for the SGR (Fig. 3) (Supplementary Data 1; Supplementary Table 3). Based on the data obtained with a mean divergence time for test lineages, the SGR distribution interval of test characters became wide between 90 and 105 Ma, and the difference in the SGR was 6.06/5.24/4.80/3.61 for the pronotum/mandible/hindwing/elytron test; the SGR distribution interval became narrow during the period of 72-74 Ma, and the difference in the SGR was 0.55/0.41/0.40/0.29 for the elytron/hindwing/mandible/pronotum test. The SGR of Scarabaeidae reached the first peak (1.52/0.75/0.53/0.47) for the pronotum/mandible/hindwing/elytron at 156 Ma (Node 16, 95% CIs from 102 to

**Fig. 1 | Morphological differentiation and ancestral feeding types of Scarabaeidae. a** The principal component analysis of the mandible/pronotum/elytron/hindwing of beetles in Scarabaeidae with different feeding types. A silhouette image of scarab is selected as the example, the pronotum in this diagram has been edited to the right only for illustrative purposes. **b** Possible ancestral feeding types of the major lineages in Scarabaeidae based on the feeding types of 19 existing subfamilies and Glaresidae in Scarabaeoidea. All test groups in Scarabaeidae were labeled for follow-up analysis, and the coprophagous/omnivorous/phytophagous feeding type at each ancestral node was marked by purple/dark blue/dark green stars, respectively.

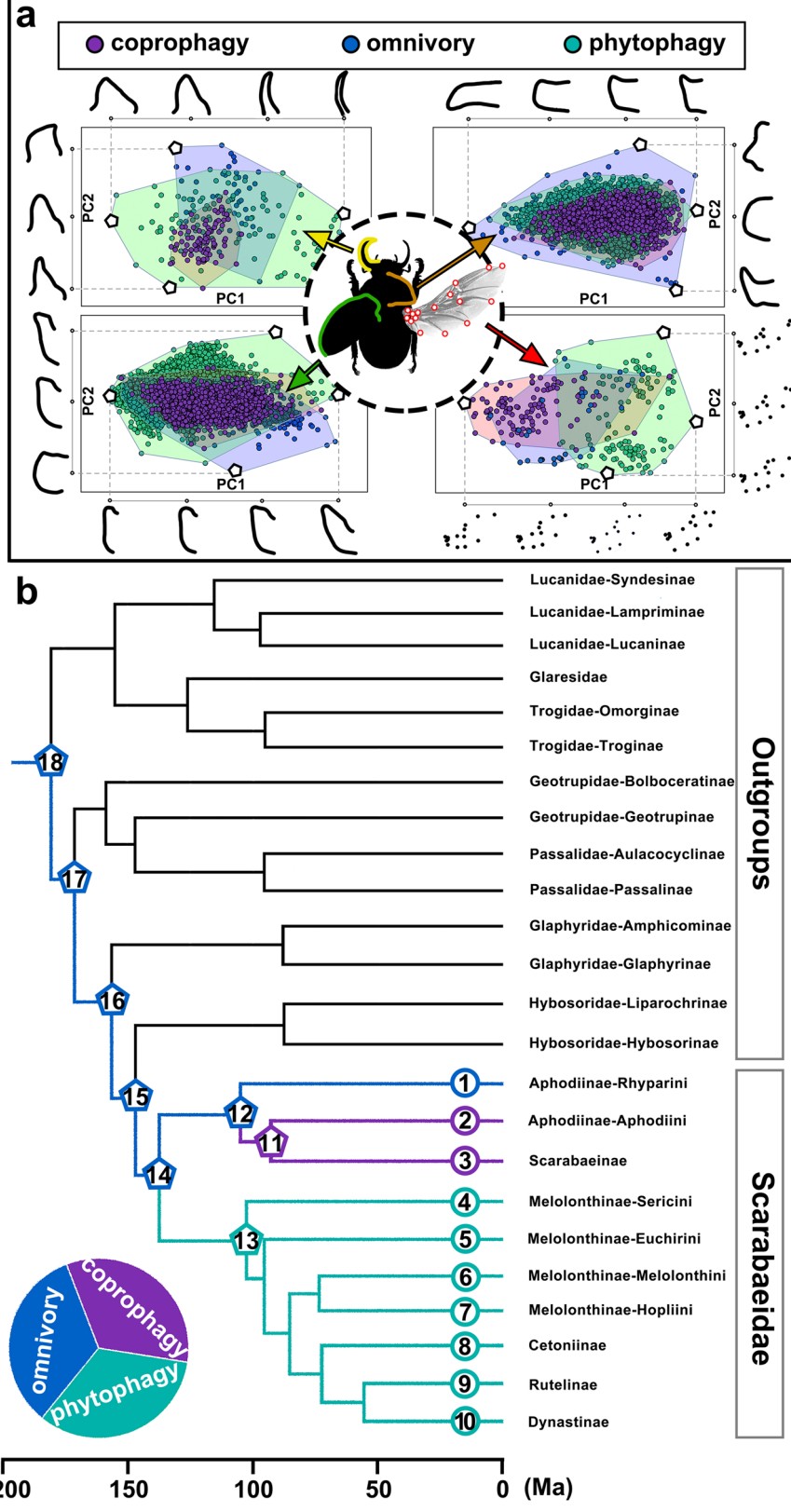

178 Ma) and the second peak ~100 Ma after differentiation of the ancestral feeding types.

For the coprophagous lineage, the SGR peaked at a mean divergence time of 105 Ma (Node 12, 95% CIs from 66 to 126 Ma), and the morphological fluctuation index of test characters at the peak point was 2.73/1.60/

1.55/0.86 for the elytron/hindwing/pronotum/mandible (see the purple dotted frame in Fig. 3). PCA showed the deformation of the elytron at the peak point in the coprophagous lineage: regarding the difference in the elytron in morphological space, the elytron of Scarabaeinae became shorter in the horizontal direction and wider in the vertical direction, while the

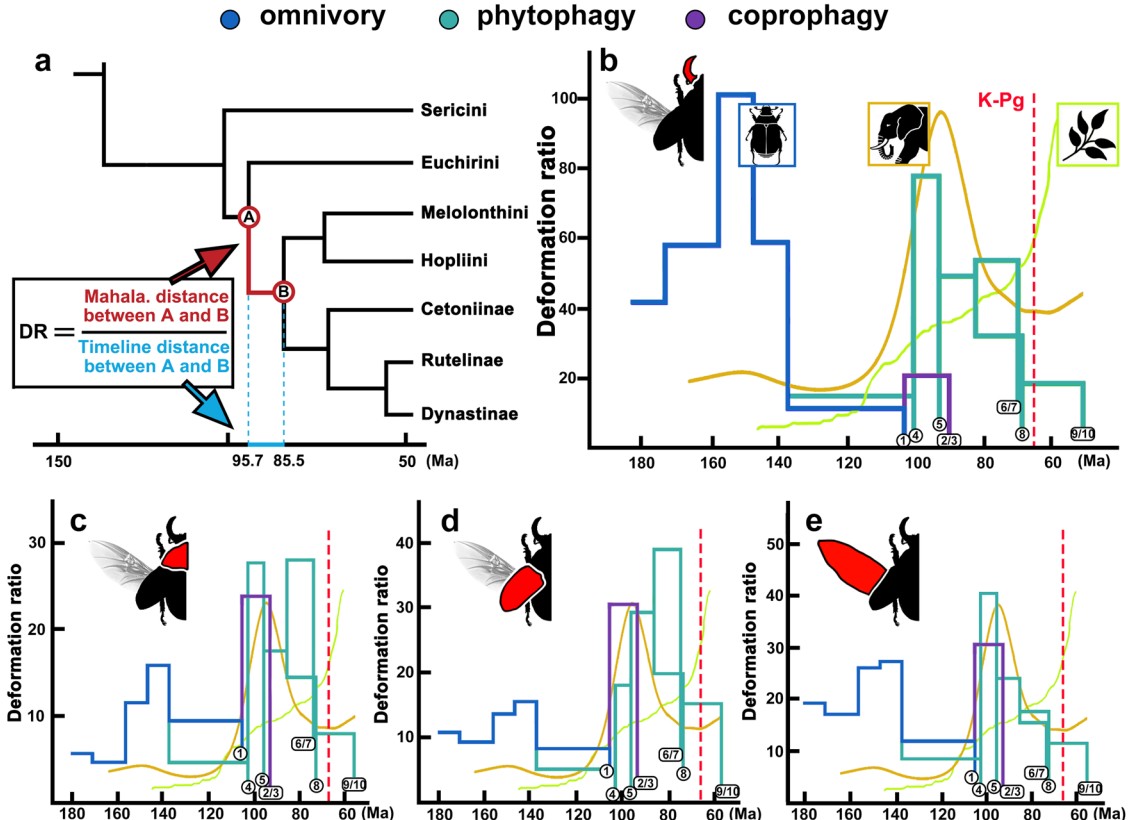

**Fig. 2 | Characters' DR in major lineages with different feeding types in Scarabaeidae. a** Diagram for obtaining the DR index based on Mahalanobis distances; **b** step line chart of the DR of the mandible; **c** step line chart of the DR of the pronotum; **d** step line chart of the DR of the elytron; **e** step line chart of the DR of the hindwing. All test groups in Scarabaeidae are labelled according to the phylogenetic tree. The solid yellow/green trendline shows the mammalian net diversification rate[42] and relative angiosperm diversity[35] without meaning in the *Y* axis direction, respectively. All test groups in Scarabaeidae were labeled for follow-up analysis, and the coprophagous/omnivorous/phytophagous feeding type at each ancestral node was marked by purple/dark blue/dark green stars, respectively.

elytron of Aphodiini showed the opposite change, becoming narrower and longer overall. CMA was conducted to quantitatively corroborate that the morphology of the elytron was significantly different among species belonging to the lineage of coprophages: the total discrimination of the groups was 98.99%, with 99.12%/98.97% discrimination rates for Ahpodiini/Scarabaeidae (Supplementary Table 1).

For the phytophagous lineage, the SGR peaked at a mean divergence time of 102 Ma (Node 13, 95% CIs from 66 to 122 Ma), and the morphological fluctuation index of test characters at the peak point was 5.13/4.34/3.90/2.61 for the pronotum/mandible/hindwing/elytron test (see the dark green dotted frame in Fig. 3). PCA of existing samples showed the deformation of the pronotum at the peak point: regarding the difference in the pronotum in morphological space, the pronotum of Melolonthinae and Rutelinae became wider in the horizontal direction and shorter in the vertical direction, while it showed the opposite pattern in the groups of Cetoniinae and Dynastinae. CMA was conducted to quantitatively corroborate that the morphology of the pronotum was different among existing species belonging to the lineage of phytophages: the total discrimination rate of the groups was 65.06%, with rates of 89.11%/72.79%/65.65%/31.51% for Cetoniinae/Dynastinae/Melolonthinae/Rutelinae (Supplementary Table 1). The SGR of phytophages gradually weakened after the mean divergence time of 95 Ma (although the SGR of the pronotum in Melolonthini reached the third peak at a mean divergence time of 85 Ma), and various taxa were differentiated at the same time.

## Discussion

Based on the reconstruction of ancestral feeding types, we revealed that the ancestor of the Scarabaeidae was primarily omnivorous at first, changing gradually into two biological groups in the early Cretaceous that were dominated by coprophages and phytophages. The morphology of scarabs with different feeding types evolved functionally and displayed differences between taxa. This pattern of ancestral dietary transition was also in line with earlier findings based on the structure of the mandible and a few gene fragments[5,21,28,29]. Simultaneously, we proposed that extant scarabs in the tribe Rhyparini, which have remained omnivorous throughout the process of feeding type differentiation of the Scarabaeidae, feed on mosses and fungi or coexist with termites[30–33]. This result was further supported by the finding that the coprophagous feeding type transition of ancestral Scarabaeinae occurred after the divergence of the common ancestor of Aphodiinae and Scarabaeinae[22].

On the basis of phylogenetics and geometric morphometrics, we assessed changes in morphological diversity on continuous spatiotemporal scales, and we revealed the relationship between the evolutionary radiation of scarabs and various selection pressures. By examining the DR and SGR of characters, we determined that the pronotum deformation was the most pronounced, with peaks in the Early and Middle Cretaceous, respectively. In the Early Cretaceous, the ancestral feeding type of Scarabaeidae was not differentiated and remained omnivory, and the reconstructed ancestral pronotum was strengthened once around the mean divergence time of 146 million years ago. We hypothesized that this was due to a sharp drop in global average temperature at that time, which led to enhancement of the ancestral excavation ability and the morphological diversity of the pronotum[34,35]. The above hypothesis was also supported by the evidence that fossil samples of scarabs from the Early Cretaceous period display excavation capabilities[36,37]. The most prominent trend of pronotum morphological change was caused by the accelerating change in the ecological niche and the increasing environmental selection pressure, which strengthened the prothoracic muscular system closely related to aspects of 'environmental

**Fig. 3 | SGR index of characters in Scarabaeidae.** A diagram for obtaining the SGR based on Mahalanobis distances is shown. The differently colored lines of best fit represent the SGR changes in the test characters on the space-time scale. The colored stars indicate the peak values of the SGR for different test characters. Principal component analysis was performed to examine the character differentiation of the coprophagous/phytophagous lineages at the SGR peak, and the capitalized first letter represents the different subfamilies (A Aphodiini, C Cetoniinae, D Dynastinae, M Melolonthinae, R Rutelinae, S Scarabaeinae). A line chart of global average temperatures during the evolutionary period of Scarabaeidae is shown by the gray line below, and the corresponding temperature can be read from the vertical axis on the right side of this figure[35].

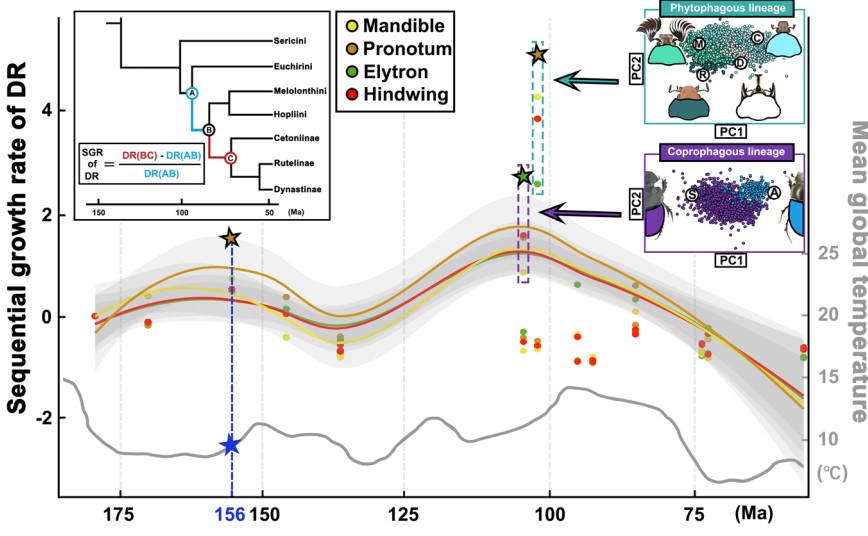

competitiveness', such as excavation and climbing[20,38]. Therefore, we speculated that the primary factor in the noticeable deformation of the pronotum during the Middle Cretaceous might have been the increasing diversity of angiosperms and the enhancement of soil nutrient release by angiosperm litter[14,39,40]. In addition, based on the greater SGR difference for all the test characters during this period, our results revealed that the scarabs' morphological changes coincided with events such as the development of mammals and angiosperms and feeding type differentiation.

We found extremely consistent deformation signals for all tested characters on the timeline, which provided strong support for the evolutionary hypothesis of the relationship between coprophagous scarabs and mammals (Fig. 2). The DR of the coprophagous ancestors in this study peaked in the Early Cretaceous. This finding was supported by the paleontological hypothesis that the key macroevolutionary events of the extant mammalian lineage occurred prior to the K-Pg boundary and that mammalian diversity typically rose rapidly again after the K-Pg boundary, driven by large increases in speciation rates[41,42]. Moreover, it corroborated that the differentiation of coprophagy characters was influenced by the development of mammalian diversity.

Studies on the evolution of coprophagy have also proposed that the diversification of the Scarabaeinae occurred as a result of savannah expansion during the Miocene, which was caused by arid and variable climates[43,44], and the dominance of Artiodactyla as the dominant feces producer[45,46]. However, since our study lacked a molecular phylogeny for the lower-order taxa and fossil evidence for coprophages in the Miocene, it was unable to reveal more about the population differentiation of Aphodiinae and Scarabaeinae during the second mammal extinction event following the Miocene. Nonetheless, combined with the test indices of functional morphological characters at the spatiotemporal scale, the morphological changes at the diversity differentiation nodes were interpreted: the variation in functional morphological characters in taxa subjected to various selection pressures is astounding. Based on the spatiotemporal analysis of the morphological characters of the above two subfamilies during the first radiation event of mammals, it was discovered that the changes in the mandible were much smaller than those in the other test characters at the differentiation node of coprophagous scarabs in the Early Cretaceous (Fig. 3), which was inconsistent with the idea that Scarabaeinae dominated dung communities because of their highly efficient and adaptable mouthparts[5,47,48]. An in-depth investigation based on PCA of Aphodiini, Rhyparini and Scarabaeinae was conducted, and we found that the morphological diversity of the elytron changed dramatically. We proposed that environmental selection pressures and ecological niche changes influenced the differentiation of the elytron in the following ways: beetles in Scarabaeinae evolved a more arched and shortened body, which was reflected in the deformation trend of lateral

expansion and longitudinal contraction of the elytron, which enhanced their digging ability and promoted their behavior of burying dung balls under the ground, characters that have been preserved by natural selection over a long period of evolution[49]. The body morphology of scarabs in Aphodiini and Rhyparini gradually evolved an elongated shape during the evolution of dwelling behavior in feces and anthills[20,31,50]. This discovery provided morphological evidence to explain the concept of niche competition between Scarabaeinae and Aphodiinae. According to molecular biology, Aphodiinae were confined to niche spaces on the periphery due to their lack of special morphological traits and the inability to transport feces of Artiodactyla[22,38]. Furthermore, by combining the results of DR and SGR, we found evidence to support the monophyly of the dung beetle clade (Scarabaeinae + Aphodiinae)[24,51]. This was demonstrated by the more consistent feedback patterns of various characters within the branches of phytophagy (Cetoniinae + Dynastinae + Melolonthinae + Rutelinae) and coprophagy (Scarabaeinae + Aphodiinae), as well as the obvious differences in feedback patterns between these two branches due to selection pressure.

Our study provided evidence that the diversity development pattern of phytophagous scarabs coincided with the rapid differentiation of angiosperms and the changing trend of global average temperature during the Cretaceous[35,52,53]. By using quantitative morphological and phylogenetic analyses, we discovered that the phytophagous scarabs' DR increased in the Middle Cretaceous. This finding validates the conclusions drawn from biogeographical and fossil data, which show that the phytophagous scarab clade originated and diversified in the Cretaceous[36,54,55]. And the deformation of the mandible and pronotum displayed a secondary peak in the Late Cretaceous, after which it gradually weakened and then remained stable in the period around the K-Pg boundary. Our study revealed the continuous evolution of biological taxa from a quantitative morphological point of view: before their explosive development in the Palaeogene[56,57], these phytophagous scarabs already responded to environmental selection pressures and formed morphological diversity in the mid-Cretaceous. We hypothesized the following situation in light of the fact that the K-Pg extinction event had no detrimental effects on the differentiation pattern of higher-order species of phytophagous scarabs (which was also observed in the global biodiversity pattern of angiosperms). Phytophagous scarabs underwent strong morphological differentiation in the mid-Cretaceous period with angiosperms' rapid differentiation[35,58], and then, the niche distribution patterns were fixed and stable differentiation and development occurred after new niche proliferation during the Palaeogene[34,59]. This hypothesis was supported by the SGR plot in this study: the continuous SGR of the morphological characters of phytophages peaked in the middle Cretaceous and then gradually decreased and stabilized after the differentiation of distinct subfamilies throughout the middle and late Cretaceous. In addition, we found that the

mandible and hindwing of the phytophages throughout this period were extremely distinct and might have been affected by the spread of new foods and populations[47,60,61].

On the basis of the reconstruction of a well-supported and dated backbone phylogeny, evolutionary models of taxa are generated by calculating the rate of clade diversification. In addition, a variety of novel methods have been created to quantify the degree to which rates of species diversification differ between lineages or in relation to character states[62-65]. Some researchers also recognize the significance of characterizing differences in feature morphology and applying them to spatial and temporal scales[1,5,66]. However, it is difficult to infer the diversification of evolutionary strategies at important nodes in the evolution of taxa, and the driving mechanism of taxonomic variety induced by selection pressures can only be hypothesized based on biological data of extant species. In this study, the relationship between evolutionary radiation and selection pressures was explored in depth by examining the deformation curves of functional features with historical ecological parameters in conjunction with the most comprehensive phylogenetic tree available (angiosperms, mammals, and global temperature). In addition, functional morphological characters on the continuous timeline were compared horizontally through the test parameters, and morphological changes were revealed by combining data on the diversity differentiation of taxa, thereby providing insight for the analysis of the evolutionary radiation of taxa at key nodes. This work provides a novel explanation for the evolutionary patterns of different biological groups and provides a foundation for research in geology and biogeography.

## Materials and methods
### Reconstruction of the ancestral feeding types of Scarabaeidae
19 subfamilies from eight families of Scarabaeoidea were selected for reconstructing the ancestral feeding types of Scarabaeidae: (1) six subfamilies in Scarabaeidae were included in the inner group; (2) 13 subfamilies from seven families of Scarabaeoidea were included in the outgroups (Fig. 1b). Test groups were divided according to the main test members or the typical representative feeding types of the test groups: (1) the omnivory: Aphodiinae-Rhyparini; Geotrupidae, Glaresidae, Hybosoridae, Trogidae; (2) the coprophagy: Aphodiinae-Aphodiini, Scarabaeinae; (3) the phytophagy: Cetoniinae, Dynastinae, Glaphyridae, Lucanidae, Melolonthinae, Passalidae, Rutelinae[5]. A phylogenetic relationship of Scarabaeoidea was revised by the published tree of 89 genes[67], then the feeding types of ancestor nodes were reconstructed through the feeding types of living taxa in Mesquite (Version: 2.72)[68].

### The selection of test characters and samples
One of the most significant biological processes in scarabs is resource partitioning, which results in significant structural modifications and adaptations for certain feeding roles or foraging behaviors[69-71]. In this work, we have chosen four typical homologous characters of scarabs that have been demonstrated to be strongly associated with feeding behaviors[21,24,72,73]: various mandibular parts determines the way the beetle handles food with different properties[5,74]; it has been demonstrated that the hindwing is crucial for increasing beetles' efficiency in their food-finding[75]. Furthermore, the pronotum and elytron are recognized as crucial components of the scarabs' body. The pronotum's morphology differs amongst feeding groups because of the different distribution of muscles attached to the prothorax, which is typically influenced by the head's movement (including feeding behavior)[75,76] and the foreleg's food handling habits (primarily coprophagous)[77,78]; the elytron is connected to the muscles involved in digging and hindfoot movement because of the way the mid/hind thorax is shaped[79], it was established that distinct phytophagous and coprophagous scarabs exhibit distinct elytron morphological changes[69,80,81].

This study was based on three datasets for increasing the morphological information obtained from each test taxa to be more representative (Supplementary Table 4), which included 9331 specimens of 6403 species for the testing of the pronotum and elytron, 250 specimens of 216 species for the mandible test, and 263 specimens of 255 species for the hindwing test[82].

Most of the specimens were deposited in the Institute of Zoology, Chinese Academy of Sciences, and the Natural History Museum London. Additional photographs of species were taken from the literatures[83-89]. The specimens were examined and dissected using a LEICA MZ 12.5 dissecting microscope, and all the photographs were taken using an Olympus EM5 (60 mm) camera. Standard dorsal images were selected for this study. To facilitate accurate representation, images were only used when the testing characters were not covered or blurry, and the images possessed adequate resolution (the smallest one was 90 pixels).

### Digitization of characters' morphological information
Three curves were extracted and resampled into 50/25/50 equally spaced semi-landmarks (SLMs) from the left contours of the mandible/pronotum/elytron through MorphoJ (Version: 1.06a)[90], for quantitative analyzing the morphology, respectively (Fig. 1a) (Supplementary Data 2). The first curve was taken from the outer contour of mandible covered by the base of the left to the base of the right, a silhouette of dorsal view of mandible was used to avoid the partial asymmetry of the left and right mandible in three-dimensional space, which could lead to instability in the results[91-93]; the second curve was collected from the middle of the anterior margin of the pronotum and end up at the middle of the posterior margin of the pronotum; the third curve started from the anterior margin of the left elytron and stopped at the end of elytron. 16 landmarks were taken from the right hindwing through MorphoJ for quantifying the structure of wing vein nodes (Supplementary Data 2), in order of numbering of landmark points: the base of ScA, the intersection of the $RA_{3+4}$ vein with the leading edge; the end of $RA_{3+4}$; the end of $RA_3$; the end of $RA_4$; the base of $RA_{1+2}$; the base of MP; the base of RP; the end of RP; the end of MP; the base of CuA; the end of CuA; the base of AA; the end of AA; the base of AP; the end of AP[81].

For the preprocessing of the mandible/pronotum/elytron dataset, all SLMs were digitized with tps-Dig (Version: 2.05)[94]. The format of data files used for morphological analysis were achieved by converting SLMs into LMs[95] in text files for the subsequent analysis: the curve number and point number for each sample were deleted, then landmark numbers were replaced by point numbers[96,97].

### Quantitative morphological analysis: morphological diversity of test characters between groups with different feeding types
In Mathematica (Version: 12.1.0.0)[95,98], the changes in the morphological diversity of the mandibles/pronotum/elytron/hindwing among the three feeding-type groups were inferred through principal component analysis (PCA) (Fig. 1a). The degree of dispersion between test groups was quantified based on confusion matrix analysis (CMA) (Supplementary Table 1).

### Acquisition of evaluation indices: DR and sequential growth rate between ancestral nodes of Scarabaeidae
In this paper, we quantified morphological changes across evolutionary nodes. The average shapes of the existing groups' test characters that were treated as the terminal taxa in the phylogenetic combined analysis were computed in MorphoJ[90]. Then, the landmarks of the test characters were entered into Mesquite[68] as a continuous matrix and linked to the topology of the phylogenetic tree[67]. The ancestral forms of all nodes were reconstructed using the traces of all characters and the landmark drawings from the modules, and the Mahalanobis distance and Euclidean distance between each pair of test groups (including all the estimated ancestral nodes and terminal existing groups) were calculated based on canonical variate analysis (CVA) in MorphoJ and Mathematica, respectively (Supplementary Data 3; Supplementary Table 2).

Then, we proposed two parameters to interpret the diversity mechanism of biological evolution at the spatiotemporal scale: (1) the DR was obtained by dividing the Mahalanobis distance and Euclidean distance of characters to the mean differentiation time of mean time between nodes (including ancestral nodes and the terminal taxa in the phylogenetic tree) (Fig. 2a), the 95% confidence interval (CI) of differentiation time between each pair of ancestral nodes was also showed for the assessment of morphological diversity; a step line chart of the DR was analyzed in SPSS

(Version: 26)[99] to evaluate the continuous changes along different branches corresponding to the same test character on the timeline. 2) The sequential growth rate (SGR) was obtained by dividing the difference value between each DR by the previous DR (Fig. 3), and a line of best fit was obtained with the geomorph (R package, Version: 4.0.5)[100] to evaluate the deformation fluctuations of different characters on the timeline (Supplementary Data 1; Supplementary Table 3).

## Statistics and reproducibility

19 subfamilies from eight families of Scarabaeoidea are used in this study, which include 9331 specimens of 6403 species for the testing of the pronotum (25 SLMs) and elytron (50 SLMs), 250 specimens of 216 species for the mandible test (50 SLMs), and 263 specimens of 255 species for the hindwing test (16 landmarks). The sample size and number of replicates for each experiment are noted in the respective section describing the experimental details.

## Reporting summary

Further information on research design is available in the Nature Portfolio Reporting Summary linked to this article.

## Data availability

All the data mentioned below can be found by the link of Dryad: https://doi.org/10.5061/dryad.qbzkh18pr

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

## Acknowledgements

We thank Norman MacLeod from Nanjing University for guiding us on geometric morphometrics and statistics analysis. We would like to thank Rongrong Shen from University of Memphis for suggestions in molecular phylogenetic test, Zhengting Zou from the Institute of Zoology, Chinese Academy of Sciences (IZAS) for evaluating the ancestral nodes' differentiation, Max Barclay and Michael Geiser from Natural History Museum London for the assistance in providing and identifying specimens, Haidong Yang from Institute of Zoology, Guangdong Academy of Sciences for the assistance in providing and identifying specimens. This work was supported by the National Natural Science Foundation of China (Nos. 32200354; 32270468; 32261143728); the China Postdoctoral Science Foundation (No. 2022M713134); the National Key R&D Program of China (Nos. 2022YFC2601200, 2023YFC2604904); the Institute of Zoology, Chinese Academy of Sciences (2023IOZ0104); the National Science & Technology Fundamental Resources Investigation Program of China (Nos. 2023FY100301, 2022FY100500, 2019FY101800); the Survey of Wildlife Resources in Key Areas of Tibet (ZL202203601); the project of the Northeast Asia Biodiversity Research Center (NABRI202203); Guizhou Science and Technology Planning Project (General support-2022-173); Rare and Endangered Species Survey Supervision and Industry Regulation Project; the First-class discipline of Prataculture Science of Ningxia University (No. NXYLXK2017A01).

## Author contributions

Y.T., M.B., and X.K. conceived the deformation indices evaluation parts and designed this study. Y.T. and Y.L. wrote the paper. Y.T. carried out geometric morphological experiments and statistics analysis. Z.T. carried out the morphological information and interpreted the data. All authors reviewed and commented on the manuscript.

## Competing interests

The authors declare no competing interests.
