## [Peer Review File · Communications Biology]

Reviewers' comments:

Reviewer #2 (Remarks to the Author):

The assumption that the "morphological characters of test scarabs were symmetrical left and right" is a rather big assumption to make. In particular, mandibles are by their nature, almost by definition asymmetric and there are many species of scarabaeoid beetles with overtly asymmetrical mandibles. I don't know how much this might affect the data, but these limitations, and resulting interpretations of the analyses will need to be qualified to indicate this possibility.

The division of scarab beetles into the three feeding categories is too coarse because many scarab beetles have other feeding habits (e.g. necrophagy, mycetophagy), and moreover, the distinctions are here demonstrably not always based upon fact because a number of the species studied are identified only to genus and hence could not have been assigned to a feeding category based upon specific published information on their feeding habits. I understand that it is very likely that for instance, within a genus, feeding habits will be uniform, but this is not necessarily so. Again, at the very least, these limitations, and the interpretations of resulting analyses will need to be suitably qualified.

Lines 131-135. The figures for the number of specimens studied for each morphological category need to be checked. e.g. the figure for pronotum and elytron seems to be incorrect.

Reviewer #3 (Remarks to the Author):

Overall, I found this manuscript to be interesting and well written, and is a good contribution to scarab systematics and evolution with broad relevance. Here are some issues that the authors should address to improve their manuscript:

- 1) The authors make the connection between morphological evolution of characters and the general feeding types of scarab groups. It would help if there were some specific statements about why the authors think that these things are strongly connected over other factors.
- 2) The authors should discuss how alternate uses for the structures they are measuring may have affected their results. For example, they have used *Fruhstorferia anthracina* in their study – a species where the males have incredibly large mandibles used to battle each other for access to breeding females. Since the mandibles in this species and some others are primarily under sexual selection pressures (not feeding pressures), how might this affect the results of this study?
- 3) The authors should address how species that have alternate feeding types within their general categories. For example, *Geotrupes* and other *Geotrupidae* are dung feeders, and many *aphodiines* are detritivores/omnivores. How does including species like this with alternate feeding affect their overall results?
- 4) In the supplemental materials, several specimens used in the study are not adequately identified – e.g., "Cetoniinae sp.", "Dynastinae sp.". The authors should make an effort to improve this as it will help make their results more meaningful.
- 5) I suggest the authors read the following paper, as is discussed many of the same evolutionary patterns for scarab beetles that their manuscript covers:
Smith, A.B.T., Hawks, D.C., and Heraty, J.M. 2006. An overview of the classification and evolution of the major scarab beetle clades (Coleoptera: Scarabaeoidea) based on preliminary molecular analyses. *Coleopterists Society Monographs*, 5: 35–46. [https://doi.org/10.1649/0010-065X\(2006\)60\[35:A00TCA\]2.0.CO;2](https://doi.org/10.1649/0010-065X(2006)60[35:A00TCA]2.0.CO;2)

Reviewer #2

1. The assumption that the "morphological characters of test scarabs were symmetrical left and right" is a rather big assumption to make. In particular, mandibles are by their nature, almost by definition asymmetric and there are many species of scarabaeoid beetles with overtly asymmetrical mandibles. I don't know how much this might affect the data, but these limitations, and resulting interpretations of the analyses will need to be qualified to indicate this possibility.

Author's response:

We thank the reviewer for the suggestions and agree with this reviewer. We discovered that the left and right mandibles of scarabs have both symmetrical and asymmetrical portions simultaneously based on our research and a number of investigations conducted by other teams. In three-dimensional space, the asymmetric portions typically exhibit bigger differences and serve particular functions. For instance, in certain rhinoceros beetles or female stag beetles, the necessity of cutting through the hard bark led to the evolution of a hard, interlocking gear-like structure at the base of the mandible, which has been shown to be extremely complex in three dimensions (Tomas et al., 2018; Wataru et al., 2019); in certain scarabs that visit flowers, the surface structure of the left and right molars differs; for instance, on the surface of the right molar, a sclerotized, curved ledge can be observed, while the left molar exhibits a corresponding ridge (Florian et al., 2015).

In order to avoid higher dimensional data calculations during the quantitative morphology process, we chose to analyze the left outer contour of the mandible silhouette in this study (please refer to the schematic diagram below). We believe that this data processing method has avoided the majority of morphological differences in 3D space, including left-right asymmetry. The left contours of the mandible have also been shown in our earlier study to exhibit significant morphological differences and changes during the evolutionary process of the three feeding groups, therefore, this character will be tested in this study to further investigate the evolution of the aforementioned groups on a transverse spatial and temporal scale.

Simultaneously, we discovered—in response to the reviewer's feedback—that the relevant content in the paper lacked correct and clear presentation, so we revised this section (see the Line 365-370 of the manuscript, please).

2. The division of scarab beetles into the three feeding categories is too coarse because many scarab beetles have other feeding habits (e.g. necrophagy, mycetophagy), and moreover, the distinctions are here demonstrably not always based upon fact because a number of the species studied are identified only to genus and hence could not have been assigned to a feeding category based upon specific published information on their feeding habits. I understand that it is very likely that for instance, within a genus, feeding habits will be uniform, but this is not necessarily so. Again, at the very least, these limitations, and the interpretations of resulting analyses will need to be suitably qualified.

Author's response:

Thank you for this insightful recommendation. We concur with the reviewer that each taxon contains unique species that, as a result of the ecological niche or functional requirements during evolution, alters their shape and habits. In sight of this, we have created a series of study plans that use a combination of quantitative morphology and phylogenesis to investigate the variety of Scarabaeidae from both macroscopic and microscopic evolutionary perspectives.

Our focus in this study is on the macroscopic evolution of the Scarabaeidae family. We chose the primary members' feeding types or the test group's typical representative feeding types to represent it in the higher category and investigated the overall evolutionary trend of the higher categories, which were represented by the study's families and subfamilies in order to uncover the spatiotemporal changes of each branch from ancestor to living taxa. For each test group, we increased the number of test samples as much as we could in order to more accurately depict the morphological divergence of the variety of these higher-categories overtime. To better understand the diversity from the perspective of microevolution, we intend to concentrate the test samples in the next study on lower categories with greater research feasibility, like the subfamily Scarabaeinae. We will also choose more test features, like the head's horn, and higher species sampling coverage (higher representation) as well as more feeding types, like predatory, omnivorous, and coprophagous.

In order to more precisely define our subsections and partially restrict our conclusions, we simultaneously modified the respective methodology and discussion parts (see the Lines 56-58, and 332–333 of the manuscript, please).

3. Lines 131-135. The figures for the number of specimens studied for each morphological category need to be checked. e.g. the figure for pronotum and elytron seems to be incorrect.

Author's response:

I appreciate your suggestion. We double-checked and updated the number. Additionally, we were able to update the number of tested species listed in the manuscript and identify some of the specimens in greater detail with the assistance of

specialists, making our dataset accessible to a wider audience (see Line 354-356 of the manuscript, and the species with red font in Supplementary_Table_2, please).

Reviewer #3

1. The authors make the connection between morphological evolution of characters and the general feeding types of scarab groups. It would help if there were some specific statements about why the authors think that these things are strongly connected over other factors.

Author's response:

We appreciate the reviewer's insightful feedback. Based on it, we have also recognized some deficiencies and have revised this section to make sure the reader is not confused about the test character selection (see the Line 340-352 of manuscript, please).

2. The authors should discuss how alternate uses for the structures they are measuring may have affected their results. For example, they have used *Fruhstorferia anthracina* in their study – a species where the males have incredibly large mandibles used to battle each other for access to breeding females. Since the mandibles in this species and some others are primarily under sexual selection pressures (not feeding pressures), how might this affect the results of this study?

Author's response:

We accept the reviewer's viewpoint and appreciate your suggestions. We also believed that there would be some variation in the results from individuals who were subject to the pressures of sexual selection, so we did not choose any pertinent species for investigation in this study. *Fruhstorferia anthracina* was simply utilized as a schematic diagram in this publication to indicate the four test characters (we chose this species whose mandible was exposed to demonstrate precisely how we mark the contour). Meanwhile, our deformation parameters analysis of the Scarabaeidae branch was unaffected by the use of the stag beetles, which were closely associated to sexual selection, only as the out group in this study. The original species diagram in this manuscript has been replaced with a silhouette diagram, which simply serves to illustrate the marking points (landmarks and semi-landmarks) and does not represent a specific species, in an effort to clear up any confusion for readers (see the Fig.1-2 on Line 133 and 154, please).

3. The authors should address how species that have alternate feeding types within their general categories. For example, Geotrupes and other Geotrupidae are dung feeders, and many Aphodiines are detritivores/omnivores. How does including species like this with alternate feeding affect their overall results?

Author's response:

Thank you for this insightful recommendation. We concur with the reviewer that each taxon contains unique species that, as a result of the ecological niche or functional requirements during evolution, alters their shape and habits. In sight of this, we have created a series of study plans that use a combination of quantitative morphology and phylogenesis to investigate the variety of Scarabaeidae from both macroscopic and microscopic evolutionary perspectives.

Our focus in this study is on the macroscopic evolution of the Scarabaeidae family. We chose the primary members' feeding types or the test group's typical representative feeding types to represent it in the higher category and investigated the overall evolutionary trend of the higher categories, which were represented by the study's families and subfamilies in order to uncover the spatiotemporal changes of each branch from ancestor to living taxa. For each test group, we increased the number of test samples as much as we could in order to more accurately depict the morphological divergence of the variety of these higher-categories overtime. To better understand the diversity from the perspective of microevolution, we intend to concentrate the test samples in the next study on lower categories with greater research feasibility, like the subfamily Scarabaeinae. We will also choose more test features, like the head's horn, and higher species sampling coverage (higher representation) as well as more feeding types, like predatory, omnivorous, and coprophagous.

In order to more precisely define our subsections and partially restrict our conclusions, we simultaneously modified the respective methodology and discussion parts (see the Lines 56-58, and 332–333 of the manuscript, please).

4. In the supplemental materials, several specimens used in the study are not adequately identified – e.g., “Cetoniinae sp.”, “Dynastinae sp.”. The authors should make an effort to improve this as it will help make their results more meaningful.

Author's response:

We appreciate your careful review and helpful recommendations. Following a thorough examination, we found all of the specimens with this naming type (e.g. Subfamily sp.). In order to obtain the names of these species, we first got in touch with Mr. Haidong Yang from the Institute of Zoology at the Guangdong Academy of Sciences. Yang provided the aforementioned data based on his thesis, ‘Geometric Morphometrics Evaluation of Three-dimensional Morphology of Scarab Hindwing Articulations and the Co-evolution Study, Xinjiang Agricultural University, 2016’ (see the species with red font in Supplementary_Table_2, please). Furthermore, we got the helps from Prof. Max Barclay and Mr. Michael Geiser of the Natural History

Museum London in order to verify the original specimens for the identified species (e.g., Passalidae-Aulacocyclus; Passalidae-Passalinae; Glaphyridae-Amphicominae; Glaphyridae-Glaphyrinae; Glaresidae; Geotrupidae-Bolboceratinae; Hybosoridae-Hybosorinae). Additionally, we double-checked the pertinent specimens from Institute of Zoology, CAS (e.g. Scarabaeidae-Aphodiinae-Rhyparini) in order to identify the above species in greater detail and to increase the availability of our dataset in this round of revisions (see the species with red font in Supplementary_Table_2, please). We express our gratitude to everybody who contributed to this round of changes, their names have been included in the manuscript's acknowledgments section.

5. I suggest the authors read the following paper, as is discussed many of the same evolutionary patterns for scarab beetles that their manuscript covers: Smith, A.B.T., Hawks, D.C., and Heraty, J.M. 2006. An overview of the classification and evolution of the major scarab beetle clades (Coleoptera: Scarabaeoidea) based on preliminary molecular analyses. *Coleopterists Society Monographs*, 5: 35–46. [https://doi.org/10.1649/0010-065X\(2006\)60\[35:AOOTCA\]2.0.CO;2](https://doi.org/10.1649/0010-065X(2006)60[35:AOOTCA]2.0.CO;2)

Author's response:

We are very grateful to the reviewer for the constructive suggestion and support for our research. Through reading the relevant literature, we have a deeper understanding of the differentiation process of the Scarabaeoidea, as well as the evolutionary history of the phytophagy and coprophagy under the family Scarabaeidae, which is more relevant to our study, and these evolutionary histories have gradually become clear with the joint efforts of different analysis methods and research perspectives of researchers in different eras. In addition, we are very excited to find that our findings are consistent with some of the conclusions and hypotheses, and that the relative study of the evolution of taxa below subfamily-category has provided us with more help in our follow-up work.

We have rewritten some of the content discussed in the relevant literature (see the Line 272-277 and 281-284 of the manuscript, please).

REVIEWERS' COMMENTS:

Reviewer #2 (Remarks to the Author):

Having read the authors' responses to the specific concerns that I raised during review, and seen the corresponding modifications and additions that they have made to the text, I am satisfied that these have now improved the manuscript to the point of making the scientific content acceptable for publication.